# Population Age Structure and Greenhouse Gas Emissions from Road Transportation: A Panel Cointegration Analysis of 21 OECD Countries

**DOI:** 10.3390/ijerph17217734

**Published:** 2020-10-22

**Authors:** Hyungwoo Lim, Jaehyeok Kim, Ha-Hyun Jo

**Affiliations:** Department of Economics, Yonsei University, 50 Yonsei-ro, Seodaemun-gu, 03722 Seoul, Korea; hyungwoo.lim0206@yonsei.ac.kr (H.L.); safin84@yonsei.ac.kr (J.K.)

**Keywords:** age structure, GHG emissions, road transportation, panel cointegration, FMOLS

## Abstract

Using panel data from 21 Organization for Economic Cooperation and Development (OECD) countries collected between 2000 and 2016, this study analyzes the effect of age structure on greenhouse gas (GHG) emissions from road transportation. Previous studies have failed to reflect the driver’s behavior patterns, especially by age group. We apply the Fully-Modified Ordinary Least Squares (FMOLS) method, including the age structure effect by reorganizing 17 age groups into a polynomial structure. The age structure exhibits an asymmetric inverted U-shaped effect on GHG emissions. Initially, people emit more GHGs as they age, and reach peak emissions in their late 20s, after which emissions fall until around the age of 70, when GHG emissions remain constant because of minimum mobility demand. Factors, such as higher income, increased vehicle ownership, and raised transport volumes increase emission rates. On the other hand, fuel transition and increased fuel price, population density, urbanization rate, and fuel economy reduce GHG emissions. Furthermore, we perform a projection of GHG emissions until 2050, and conclude that the effect of age structure is limited because of the minimum mobility demand of the elderly. We conclude that various policy measures, such as increased fuel economy and urbanization, must be considered in order to achieve sustainable transport

## 1. Introduction

In 2015, the Paris Agreement was adopted to mitigate greenhouse gas (GHG) emissions and restrict global warming below 2 °C. One hundred and ninety-three countries submitted Nationally Determined Contributions (NDCs), aimed at stemming climate change. Among these contributions, 81% mentioned transportation as a relevant mitigation area, and 60% suggested transport-related mitigation measures [1]. Even though the importance of transportation relative to other polluting sectors is not regularly highlighted, transportation is one of the largest producers of greenhouse gas (GHG) emissions. According to the Intergovernmental Panel on Climate Change (IPCC), transportation produced 7 Gigaton CO_2_ equivalent (GtCO_2_eq) in 2010, which accounted for almost 23% of all energy-related CO_2_ emissions [2]. Furthermore, the IPCC’s 5th Assessment Report predicts that, without aggressive mitigation policies, transportation’s CO_2_ emissions could reach 12 Gt by 2050.

The problem of GHG emissions from transportation is especially serious in developed countries. In 2010, transportation’s share of total emissions produced by high-income countries was 22%, whereas in low- and middle-income countries, it was 3% and 8%, respectively [2]. Hence, greater focus should be placed on developed countries, such as Organization for Economic Cooperation and Development (OECD), when attempting to reduce GHG emissions from transportation.

The good news is that transportation has significant potential to reduce GHG emissions. From 1990 to 2010, the emission intensity of transport decreased by 58%, and is expected to decrease further [2]. Moreover, with expansion in the supply of electric vehicles (EVs) the possibility of a fuel transition is growing. Global EV sales accounted for only 0.6% of total vehicle sales in 2015, but this has since grown to 3% by early 2020 [3]. In the near future, the use of EVs is expected to become more common, and public transportation systems are expected to switch to less polluting alternatives, such as compressed natural gas (CNG) buses. 

Until now, the relevant literature has focused on the Stochastic Impacts by Regression on Population, Affluence and Technology (STIRPAT) model [4]. This model uses income, population, and fuel economy variables to explain the GHG emissions from transportation. However, this approach cannot reflect the behavioral patterns that depend on age and how these affect transport demand. Since the actual transport demand and GHG emissions are determined by people’s behavioral patterns, neglecting to consider population age structures may cause relevant trends to be missed.

Some recent studies have analyzed the link between age structure and GHG emissions from transportation [5,6,7]. From the survey data collected, travel distance is found to diminish as people age, with over-70s traveling less than half the distance of people in their 50s [8]. However, the preference for personal mobility is higher for the elderly. The proportion choosing personal vehicles was almost 80% for those aged 70 and over [8]. Moreover, older people reported that they want to engage in more activities and wish to travel more. This demonstrates that aging societies can effectively reduce GHG emissions from transportation. However, since the elderly prefer personal vehicles and wish to satisfy a minimum mobility demand, the reduction caused by aging could be somewhat limited.

This study builds upon previous studies’ results in three ways. First, we develop previous studies which did not account for the effects of age structures by dividing people into 17 age groups and analyzing the effect of age on GHG emissions from road transport in 21 OECD countries. This represents an expansion on previous studies that only considered the proportion of over 65s, or divided into only two or three age groups [9,10,11]—an approach that only provides partial insight into the effect of age structures. Combining age groups too broadly and analyzing the behavioral patterns of large groups can miss important differences (e.g., people in their 80s may act differently to those in their 60s). By using the population polynomial method suggested by Fair and Dominguez [12], we analyze the detailed behavioral differences across all age groups. We also identify the age group responsible for the highest rates of GHG emissions and the turning point after which GHG emissions reduce. To our knowledge, this study is the first time that such a method has been applied to transportation.

Second, to examine the long-run relationship between age structure and emissions from road transport, we use a panel cointegration regression model. Age structures change slowly and affect driving patterns in a long-run. That is, we need to analyze the long-term effects of the age structure. While the literature generally analyzed this relationship using short-run static models [12,13,14], we improve on this approach by considering long-run relationships using cointegration tests and modeling them by the Fully-Modified Ordinary Least Squares (FMOLS) method [15].

Third, we consider a greater number of the determinants of GHG emissions from transportation, including factors suggested by the previous literature as well as fuel economy and the fuel transition effect. Due to the difficulty of obtaining data, the fuel efficiency variable is usually considered via a single country or an European Union (EU)-wide analysis [16,17,18]. We build the fuel economy variable combining data from the Odyssee-Mure Project and the International Council on Clean Transport (ICCT) dataset. We also consider recent fuel transition trends by using the proportion of unconventional fuel used in road transportation. Lastly, in order to test the age structure effect over the long term, we produce a projection of GHG emissions from road transportation up to 2050. From this analysis, we can measure the effect of age structure change and identify which countries should experience significant declines in GHG emissions. This projection can form a basis for future benchmarking and could identify important policy implications. 

The rest of this paper is organized as follows. Section 2 reviews the literature and presents our study’s contribution. Section 3 introduces the data used and the methods for estimating the age structure effect. Section 4 presents the empirical results and the projection of GHG emissions from road transportation. Section 5 presents discussion topics. Section 6 concludes.

## 2. Literature Review

### 2.1. Determinants of CO_2_ Emissions from Transportation

In this section, we focus on recent studies that analyzed the determinants of CO_2_ emissions from transportation. To summarize, the major factors shown by previous studies to influence transport emissions are income, fuel price, urbanization (population density), fuel economy, transport demand, and fuel choice (fuel transition). 

The most influential factor is economic activity. Income (economic activity) is commonly identified as the most significant factor by a few studies that adopted index decomposition approaches [19,20,21]. Economic activity stimulates mobility demand, which affects the emission. Moreover, high fossil fuel prices are found to reduce GHG emissions by reducing fossil fuel consumption [16,17,22], such that increasing fuel prices has been shown to be an effective means of reducing GHG emissions [17]. Urbanization and population density also affect GHG emissions, but the direction of impact differs between developing and developed countries [23,24,25,26,27,28]. As a country becomes more urbanized, producers and consumers become geographically separated, which increases the demand for transport. Jones [23] analyzed 59 developing countries and found that a 10% increase in urbanization rates increases energy consumption per capita by 4.5%. Furthermore, according to York, Rosa, and Dietz’s [24] analysis of 137 countries, urbanization causes linear increases in GHG emissions. Yet, by contrast, in developed countries, urbanization leads to improvement in public infrastructure (such as public transportation) and reduces CO_2_ emissions. Newman and Kenworthy [25] examined 32 high-income cities and found that urban density lowers energy consumption in the transportation sector, while Hilton and Levenson [26] analyzed population density and gasoline consumption in 48 countries and found a significantly negative relationship. Liddle [27] tested the relationship between the population density and per capita road transportation energy consumption of OECD countries and found that high population density lowers energy consumption. However, other recent studies have shown that, in fully urbanized countries, urbanization has little influence on GHG emissions [9,28], and Liddle and Lung [9] demonstrated that urbanization rates do not directly affect CO_2_ emissions in the transportation sector.

Fuel economy has also been identified as an important factor in reducing energy consumption and GHG emissions in road transportation [16,18]. However, some studies have highlighted the “rebound effect” of efficiency gains—that increased fuel efficiency can lead to an increase in automobile use, which can offset the initial reduction [17,29]. Indeed, Clerides and Zachariadis [29] showed that the rebound effect could even increase GHG emissions, whilst Chai et al. [17] argued that, in China’s case, although the country’s fuel efficiency policy improved actual fuel economy, the resulting emissions reduction was not significant, citing the impact of the rebound effect. Additionally, the number of vehicles and the volume of transport are possible factors in the volume of GHGs emitted [17,30,31,32]. These variables represent overall transport demand, where as demand for transportation increases, the energy consumption of road transportation increases, which in turn leads to higher GHG emissions.

Recently, fuel transition has been receiving more attention as a prominent GHG emission reduction measure [33,34]. According to Oshiro and Masui [33], the usage rate of low-pollution vehicles in Japan is expected to increase by more than 60% by 2050, which will reduce GHG emissions from transportation by 81%, from 1990 levels. Transport and Environment [34] conducted a life-cycle analysis on the GHG emissions of electric vehicles. Assuming worst-case scenarios—in which batteries are produced in China and charged in Poland, which has a high rate of coal-fired power generation—the study found that electric cars would still emit 22% less GHGs than diesel cars.

### 2.2. Age Structure and the GHG Emissions of Road Transportation

Recently, a number of studies have focused on the effect of age structure on GHG emissions via behavioral pattern [9,10,11,35,36,37]. Kim, Lim, and Jo [35] analyzed the effect of young and old populations on CO_2_ emissions in Korea, and found that the youth are high emitters and that the elderly are relatively low emitters. Schmöcker et al. [10] and Paez et al. [11] revealed that age has a significant impact on travel propensity, demonstrating that travel propensity reaches its maximum between the ages of 34 and 50, and then drops sharply after 65. Thus, the total demand for transportation decreases as the population passes middle age. Okada [36] examined 25 OECD countries to analyze the effect of age structure on per capita CO_2_ emissions from road transportation and found the existence of an inverse U-shaped relationship between the elderly population ratio and GHG emissions from road transportation. The point at which the effect reverses was when the elderly ratio reaches 16%, indicating that carbon emissions decrease as society ages. Liddle and Lung [9] and Liddle [37] considered the relationship between age structure and CO_2_ emissions using the STIRPAT model. Liddle and Lung [9] split ages into three groups: 20–34, 35–64, and 65–79. According to their analysis, young people (aged 20–34) increase CO_2_ emissions but the others have a negative impact on emissions.

Some studies also mention technological issues related to the driving pattern of the elderly [6,38]. Using trip survey data, Rosenbloom [6] found that older people use public transportation less often and emit more air pollutants due to “cold starts,” as the majority of emissions occur as the engine and the catalyst converter heat up. Thus, even when the elderly travel short distances, their preference for private transportation leads to relatively high GHG emissions. Weilenmann, Favez, and Alvarez [38] also argued that additional pollutants can be emitted because catalyst converters are not used efficiently during cold starts. 

Until now, most previous studies have only considered certain age groups when analyzing age structure [9,10,11,35,36,37]. Due to correlation between age groups, it can be difficult to consider all age groups within models. However, incorporating only a few large age groups could not capture a detailed overall pattern of GHG emissions caused by different age groups. Moreover, even people aged over 65 could have different effects on emissions depending on their specific age; For instance, people aged 65–69 might act differently to those that are over 80. Furthermore, transport emission patterns can vary depending on environmental factors related to the age. According to Räty [39], Swedish people born before 1945 consume less energy than those who were born later. However, in Germany, people born before 1945 consumed more energy than those born after 1945. This tells us that, alongside age structure, a number of environmental factors also affect energy consumption patterns, which then impact GHG emissions.

In addition, older people have at least a certain amount of minimum mobility demands, with the elderly preferring to use cars to satisfy this demand [40]. Baby boomers value driving, seeing it as a necessary aspect of their independence and wellbeing [41], but perhaps also because baby boomers first learned to drive in a time of road network expansion. As a result, people preferred using their own vehicles for personal mobility [40,42]. This is supported by driving license data: The Federal Highway Administration in the U.S. announced that 90% of the 65-year-old male population and 80% of the female population have licenses, and that this ratio is gradually increasing [40]. 

## 3. Materials and Methods

### 3.1. Modeling the Age Structure Effect

We choose the factors known to affect GHG emissions in the transportation sector, concentrating especially on age structure effect. We set up a baseline model as follows:(1)$$\begin{array}{cc} e_{it}=\mu_{i}+\beta_{1}A_{it} & +\beta_{2}Y_{it}+\beta_{3}FP_{it}+\beta_{4}VO_{it}+\beta_{5}PD_{it}+\beta_{6}UR_{it}+\beta_{7}FE_{it}+\beta_{8}TV_{it}\\ & +\beta_{9}FT_{it}+\epsilon_{it} \end{array}$$
where $e_{it}$ is per capita GHG emissions from road transportation for country *i* at time *t*, $\mu_{i}$ is country individual heterogeneity, $A_{it}$ is age structure, $Y_{it}$ is income, $FP_{it}$ is fuel price, $VO_{it}$ is vehicle ownership, $PD_{it}$ is population density, $UR_{it}$ is urbanization, $FE_{it}$ is fuel economy, $TV_{it}$ is transport volume, and $FT_{it}$ is the fuel transition effect. 

While previous studies have only considered certain age groups in their models, we apply the modeling scheme for the whole age structure suggested by Fair and Dominguez [12]. In this way, we can analyze the effects of all age groups. The following derivation is a process of modeling polynomial age structure variables. We derive a representative agent’s emission model based on the individual’s emission.

For simplicity, we assume a single national model and group variables of the baseline model into two vectors: personal factors ($X_{ht}$) and macro factors ($W_{t}$). Personal factors include income, fuel price, and vehicle ownership. Macro factors reflect the transport environment and include population density, urbanization, fuel economy, transport volume, and the fuel transition effect. Thus, CO_2_ emissions from an individual economic agent (*h*) can be modeled as follows:(2)$$E_{ht}=\delta +\sum_{j=1}^{J}\alpha_{j}AG_{ht}^{j}+X_{ht}\beta +W_{t}\gamma +\epsilon_{ht}$$
where $E_{ht}$ is the GHG emissions of agent *h* in period *t* and $AG_{ht}^{j}$ is the *j*th age group dummy variable for agent *h* in period *t*. If agent *h* belongs to the *j*th group in *t*, $AG_{ht}^{j}$ has a value of 1; otherwise, it is 0. 

Our goal is to build a representative agent per capita model. Thus, by summing Equation (2) for the entire population ($N_{t}$), we produce the following national model:(3)$$E_{t}=\delta N_{t}+\sum_{j=1}^{J}\alpha_{j}N_{t}^{j}+X_{t}\beta +N_{t}W_{t}\gamma +E_{t}$$
where $E_{t}$ is GHG emissions at the national level and $N_{t}^{j}$ represents the number of people in group *j* in period *t*. $X_{t}$ is the summation of the individual factors over population (i.e., $X_{t}=\sum_{h}^{}X_{ht}$).

Dividing both sides of Equation (3) by total population ($N_{t}$) yields the following representative agent model:(4)$$e_{t}=\delta +\sum_{j=1}^{J}\alpha_{j}p_{t}^{j}+x_{t}\beta +W_{t}\gamma +\epsilon_{t}$$
where $p_{t}^{j}$ represents the ratio of people in group *j* in period *t*, and $e_{t}$ and $x_{t}$ are the per capita GHG emissions and individual factors, respectively.

In Equation (4), $\alpha_{j}$ denotes the effect of age groups. However, three problems arise if we use the above equation as specified in Juselius and Takáts [13]. First, as the number of groups increases, the number of parameters to be estimated also increases. Hence, the accuracy of the estimates would be reduced. Second, the more we divide the population into small groups, the higher the correlation between the groups. For instance, a group of ages 0–9 can, for example, be divided into two groups: ages 0–4 and 5–9. This adjustment gives us more information about the effect of age, but the correlation between groups increases as groups are divided, forming a direct tradeoff. Finally, since the sum of all groups is 1, it leads to perfect multicollinearity with a constant term. Hence, it is not possible to incorporate all groups.

To solve these problems, we restrict coefficient $\alpha_{j}$ to lie on a *k*th degree polynomial function suggested by Fair and Dominguez [12] and Juselius and Takáts [13]. In this way, the number of parameters to be estimated decreases, which has the advantage of enhancing the degree of freedom. Moreover, this method ensures that the coefficient between groups does not change excessively.
(5)$$\alpha_{j}=\sum_{k=0}^{K}\;\gamma_{k}j^{k}$$
where $\sum_{j=1}^{J}\alpha_{j}=0$.

Combining Equations (4) and (5) with the restriction ($\sum_{j=1}^{J}\alpha_{j}=0$) yields [14] (for a detailed derivation, see Appendix B): (6)$$e_{t}=\delta +\sum_{k=1}^{K}\gamma_{k}z_{t}^{k}+x_{t}\beta +W_{t}\gamma +\epsilon_{t}$$
where $z_{t}^{k}=\sum_{j=1}^{J}j^{k}p_{t}^{j}-\frac{1}{J}\sum_{j=1}^{J}j^{k}\sum_{j=1}^{J}p_{t}^{j}$.


After setting up a single country model, we can extend the above model into a multinational OECD panel model. Accordingly, the baseline panel model takes the following form:(7)$$\begin{array}{cc} e_{it}=\mu_{i}+\sum_{k=1}^{K}\gamma_{k}z_{it}^{k} & +\beta_{2}Y_{it}+\beta_{3}FP_{it}+\beta_{4}VO_{it}+\beta_{5}PD_{it}+\beta_{6}UR_{it}+\beta_{7}FE_{it}+\beta_{8}TV_{it}\\ & +\beta_{9}FT_{it}+\epsilon_{it} \end{array}$$

In this study, our dataset includes 21 OECD countries measured from 2000 to 2016, with age data reformatted into 17 age groups. The main purpose of this polynomial method is to include detailed age structure information while not producing too many parameters. Hence, we use the most disaggregated age groups feasibly possible, following the convention of the World Bank and the OECD in dividing ages into five-year age intervals (i.e., 0–4, 5–9,…, 70–74, 75–79, and over 80) [13]. Therefore, *J* is 17 for Equations (2)–(5).

### 3.2. Data

In this paper, we analyze 21 OECD countries using data from 2000 to 2016 (country list is provided in Appendix A), since 2016 is the last year that provides sufficient transport-related data. Table 1 describes this data. Among variables, *PR1* to *PR17* represent the proportion of age groups to total population. For instance, *PR1* refers to the age group of 0- to 4-year-olds and PR2 refers to 5- to 9-year-olds. The *z* variable in Equation (7) is generated from *PR* variables.

Fuel price (*FP*) is calculated by the weighted average of real gasoline and diesel prices, and the proportion of gasoline and diesel consumption in road transportation as a weighting. This measure seeks to reflect the price that an economic agent actually faces. Fuel economy (*FE*) is the average of all registered vehicles in the country. Since there is a lack of data for *FE*, we generate this variable by combing two datasets: Odyssee-Mure and ICCT data.

Figure 1 displays the age structure and average GHG emissions from road transportation since 2000. Panel A shows that the dependency rate of over 65s has rapidly increased, from less than 23% in 2000 to 30% in 2016. At the same time, the dependency rate of the youth has steadily decreased. Further, Panel A shows that the average GHG emissions from road transportation reached its peak in 2007 and decreased thereafter. Panel B of Figure 1 shows population proportions in OECD countries. The proportion of people aged over 80 in OECD countries has increased rapidly, from 3.4% in 2000 to 5.2% in 2016. Over the same period, the percentage of people aged in their 20s and 30s decreased from 14% and 15% in 2000 to 12% and 13% in 2016, respectively.

Table 2 provides descriptive statistics for all variables, except the proportion of age groups, for the entire period. Average per capita emissions from road transportation is 2.1 Millions of tonnes of oil equivalent (Mtoe) per year. The U.S. is the heaviest polluter and the least polluting countries are Poland and Hungary. 

## 4. Results

### 4.1. Panel Unit Root and Cointegration Tests

Before estimating Equation (7), we perform panel unit root tests and a cointegration test to check the stationarity of variables. We apply several unit root tests, which are commonly used for unbalanced panels: Im-Pesaran-Shin (IPS), augmented Dickey-Fuller (ADF), Phillips and Perron (PP), and cross-sectionally augmented IPS (CIPS) tests [43,44]. Based on the test results, we cannot reject the null hypothesis of a unit root for any variable (The results of the panel unit root and cointegration tests are summarized in Appendix A). For income and transport volumes, we can reject the null hypothesis in some tests but only at the 10% significance level. Therefore, we regard all variables as I (1) and test the cointegration relationship.

Since all variables are integrated of order one (I(1)), we perform the Kao cointegration test to detect the long-run relationship between the variables [45]. *z* variables, which describe the age structure, suffer from high linear correlation, with results greater than 0.9. To avoid possible multicollinearity, we organize the explanatory variables into three groups: Group 1 includes all explanatory variables and *z1*, Group 2 replaces *z1* with *z2*, and Group 3 replaces *z2* in Group 2 with *z3* (That is, Group 1 includes *z1* and the other variables (*E*, *Y*, *FP*, *VO*, *PD*, *UR*, *FE*, *TV*, and *FT*). Group 2 includes *z2* and the same other variables (*E*, *Y*, *FP*, *VO*, *PD*, *UR*, *FE*, *TV*, and *FT*). Finally, Group 3 includes *z3* and the same other variables.). That is, we confirm that the cointegration relation is maintained by changing the z variables [46]. The test result shows that all statistics are significant at a minimum level of 5%. Hence, we conclude that the null hypothesis of no cointegration can be rejected. Thus, we account for the long-run relationship among the variables and estimate the cointegration regression model.

### 4.2. Panel Cointegration Regression: Fully-Modified Ordinary Least Squares 

Given the existence of a cointegration relationship, we apply a panel FMOLS method to estimate the model [15]. To consider heterogenous variance, we apply the sandwich standard error suggested by Mark and Sul [47]. We express all the variables as natural logarithms, except the ratio variables, such as urbanization and the fuel transition ratio.

We test four baseline models. Model (1) is a benchmark model that includes all the determinants of GHG emissions from road transportation suggested by previous studies. We choose the optimal polynomial structure based on Wald test statistics of the cubic and quartic polynomials, following Arnott and Chaves [48]. Model (2) is similar to Model (1) except for the polynomial structure. We shrink the polynomial to the second degree to check the robustness of the model. Models (3) and (4) exclude some of the variables from the benchmark models. Model (3) excludes transport volume, because it has some missing values, and fuel transition. Model (4) uses only the factors related to personal transport use (Gender may affect individual patterns; however, it may not have much effect on the nation’s overall emissions. As a result, when we added the female ratio variable to the model, we found it insignificant at the 10% significance level.).

We are aware that cultural differences may affect GHG emission patterns of the countries. However, by applying the FMOLS method, we incorporate individual heterogeneity in estimating the model. From this process, cultural differences and other country-specific effects are incorporated in the estimation. 

The estimation results are provided in Table 3. The relationships identified are broadly in line with the results of previous studies. Based on the results of the benchmark model, income (*Y*) is the most influential variable in personal transport use. A 1% increase in income leads to a 0.6% rise in GHG emissions. Next, vehicle ownership (*VO*) also has a significant effect, with a 1% increase in the number of vehicles owned increasing emissions from road transportation by 0.25%. Fuel price (*FP*) has a negative effect on emissions, but the magnitude is relatively small. The results are quite stable from Model (1) through to Model (4). 

The results of the traffic environment are in line with the findings of previous studies, which identified that increased population density and urbanization lead to improved transportation infrastructure in advanced countries, and that this reduces GHG emissions from road transportation [25,38]. We also find that population density (*PD*) and urbanization (*UR*) reduce emissions significantly, while improvements in fuel economy (*FE*) also reduce GHGs from road transportation, but the effect is relatively small. This accords with previous studies, which have also indicated that the effect of improved fuel economy is limited due to the rebound effect of fuel use [17,29]. That is, improved fuel economy might reduce GHG emissions for individual trips, but this can also induce additional transport use. Hence, fuel use will also increase when fuel economy increases. Transport volume (*TV*) is found to significantly increase emissions, with a 1% increase in volume increasing emissions by 0.11%. Finally, fuel transition (*FT*)—i.e., replacing traditional fossil fuels with new low-carbon fuels—reduces GHG emissions significantly. For the age structure effect, we conclude that a polynomial of the third degree is the best choice based on the Wald test result (the Wald test results of Models (3) and (4) show that a higher polynomial structure is possible. However, as we expand the structure to the 4th and 5th polynomials, the results of the models remain robust. The results are provided on request). In addition, the F-test results for age structure are statistically significant.

Figure 2 displays the implied coefficient that links the size of each five-year demographic age group. All values are normalized to zero, so a value below zero implies a negative effect on emissions and a value above zero implies a positive effect. The graph exhibits an inverted U-shaped relationship between age and emissions. However, this shape is not symmetrical. There is a steep increase up to the age of 25–29, which falls thereafter. After the age of 70–74, little change is experienced. 

More specifically, until the age of 5–9, people have a low rate of GHG emissions, but as they get older, they emit more GHGs. The population in their late 20s (25–29 years old) emit GHGs at the highest rate, such that a 1% higher concentration of the age group increases GHG emissions by 0.53%. This can be attributed to the fact that, in their late 20s, people acquire driving licenses and begin their social lives. Thus, the demand for mobility is relatively high. After their 20s, people start to emit less GHGs. Until the age of around 50, the implied coefficient of age structure declines but still has a positively impacts emissions. The population over the age of 50 have lower GHG emissions, with the GHG reduction rate decreasing in intensity after the age of 70, and eventually stagnating. This phenomenon is consistent with the literature, which recognizes that baby boomers and older people have specific mobility demands and prefer driving themselves [39,41]. Hence, we can infer that there is a minimum mobility demand regardless of age, and this restricts GHG reduction. 

Figure 3 displays and compares the age effect of Models (1)–(4). Panel A compares between Model (2) and the benchmark, showing the difference between the polynomial structures of age effect. Panel B shows the difference between Models (3) and (4) results, which differ in terms of model specification. In all models, populations in their late 20 s are the highest emitters of GHGs. Moreover, the ages at which people shift between being above average emitters and below average emitters are similar to those of Model (1). All four models show people becoming above average GHG emitters at around the age of 10 and below average emitters in their 50 s.

In Panel A, Model (2) identifies the peak of the age effect as the late 20s and shows the slow decline of GHG emissions after this age, which then picks up pace from around the age of 50. However, a polynomial of the second order cannot depict the GHG reduction for ages 0–9 and stagnant phenomenon at the age of over 70s. Models (3) and (4) of Panel B demonstrate that the exclusion of some of the explanatory variables can exaggerate the effects of the age structure. However, the age at which the effect reverses remains roughly the same as in the benchmark model. Furthermore, the age group with the greatest GHG-producing effect (late 20s) remains the same, while the effect of GHG reduction also weakens after the age of 70 in Models (3) and (4).

### 4.3. Robustness Check

This section considers the robustness of the benchmark model, analyzing the model with alternative variables to reflect age structure. Previous studies have frequently used the ratio of the youth (and the elderly) or dependency rate to capture the age structure effect. However, as population ratios and dependency ratios satisfy the cointegration relationship with the other explanatory variables, we estimate other FMOLS models.

Table 4 summarizes the results of the robustness check. Robust (1) uses the share of the total population aged between 0 and 14 (PR0014) and over 65 (PR65). The estimates show that a larger share of people aged over 65 reduces GHG emissions. However, the youth share does not affect GHG emissions. The estimate has a negative value but is not statistically significant. Robust (2) uses the dependency rate of the population aged 0–14 (DEPR0014) and over 65 (DEPR65). Both variables are statistically significant and are associated with lower emissions rates. Furthermore, the dependency rate of the elderly has a roughly four times greater effect on emissions than the youth dependency rate. The results are therefore similar to the results of the models in Section 4.2. Hence, we conclude that an increased proportion of old age people reduces GHG emissions from road transportation.

### 4.4. Projecting GHG Emissions Using the Age Structure Outlook

These results naturally lead to the question of to what extent the change in age structure will affect GHG emissions in road transportation. Figure 4 shows countries’ youth and elderly dependency rates in 2017 and projections for 2050. These dependency projections show Greece and Japan as the most dependent, with a rate of 86%, with South Korea surging from 31% in 2017 to 84% in 2050. At the other end, Sweden’s rate is expected to rise only modestly, from 51% in 2017 to 55% in 2050.

In order to identify the effect of age structure change on GHG emissions in road transportation, we produce a projection of GHG emissions up to 2050 using a benchmark model. We apply the population age structure outlook data provided by the OECD Labor Force Statistics and assume that other factors (income, urbanization rate, fuel economy, etc.) remain stable from 2017. Table 5 summarizes these forecasts, where GHG emissions in 2010 and 2017 are the actual values of per capita GHG emissions from road transportation, while those in 2030 and 2050 are forecasted values.

As the dependency rate increases, the per capita GHG emissions decrease in all OECD countries, but at an average of 6.9% from 2017 to 2050. This shows that the impact of age structure change is not significant over the long term as the annual emissions reduction caused by age structure change is only 0.2% on average. Hence, we can conclude that even though age structure affects GHG emissions from road transportation, the effect of age structure alone is not sufficient to achieve the global warming goal.

However, it is interesting to see that remarkable drops are predicted for some countries. The country with the greatest effect is Greece, where a more than 20% fall in emissions from road transportation is expected. South Korea is also expected to reduce GHG emissions by 18% due to rapid population aging. On the other hand, the reduction rate is expected to be much lower in other countries. For some EU countries, such as Belgium, France, Denmark, and Sweden, emissions are expected to decline by only about 3% from age structure change, whilst the projected emissions change for the U.S. is only 4.5%. These results show that whilst an aging society should be expected to change transport demand patterns and reduce GHG emissions from road transportation across countries, the size of the effect will vary depending on the country’s demographic structure. Furthermore, the overall effect is expected to be small when considering the length of the projection period. From this analysis, we can conclude that the reduction effect from age structure will be limited and that aging societies alone cannot achieve sustainable transportation.

## 5. Discussion

Reducing GHGs emissions from road transportation is a goal shared by many countries. The results of this study confirm that changes in the age structure can contribute to reducing GHG emissions. However, the positive effects of age structure change are found to be limited. According to the UK Department for Transport [49], the per capita movement decreases after their 70s but the demand for shopping increases. In addition, the elderly have a strong preference for personal cars in developed countries [8]. These patterns imply that structural population change cannot significantly reduce emissions from road transport. Our projection shows that the annual reduction in GHGs deriving from age structure changes in OECD countries is only 0.2% on average. This level of reduction is not sufficient to achieve the IPCC’s 2 °C goal on its own, such that GHG will not occur without direct behavioral changes affected by other policy measures. That is, an aging society is not a solution for achieving sustainability.

Given this, we must ask what policies should be implemented to achieve the 2 °C goal. Our empirical results suggest that a fuel price tax, fuel economy improvements, and fuel transition can be effective tools in reducing GHG emissions. The results of Model (1) show that a 1% increase in fuel price lowers CO_2_ emissions by 0.05%, whilst improvements in fuel efficiency and fuel transition also reduce emissions. Thus, to make transportation sustainable, we must implement a number of complementary measures to reduce transport emissions. Notably, our results suggest that enhancing fuel efficiency is the most effective policy measure.

## 6. Conclusions

This study analyzes the effect of age structure on GHG emissions from road transportation. Using panel data collected from 21 OECD countries between 2000 and 2016, we produce a polynomial age group structure in the form suggested by the OECD [1]. We find that the relationship between age and GHG emissions exhibits an inverted U-shape. However, the effect is not found to be symmetrical. Rates of GHG emissions grow from birth until the late 20s, after which people emit less and less GHGs. From the mid-50s, people emit significantly lower rates of GHGs and it is only after the age of 70 that GHG reduction stagnates.

In addition, personal factors, such as income, fuel prices, and vehicle ownership are found to have significant effects on GHG emissions from road transport, as does the macro traffic environment, which includes population density, urbanization, transport volume, and fuel transition factors. Overall, the results agree with the findings of previous empirical studies in that income, vehicle ownership, and transport volume increase GHG emissions. However, enhanced fuel transition and increased fuel prices, population density, urbanization, and fuel efficiency reduce emissions. These results hold true even when we consider other age structure variables, such as the population ratio and dependency rate.

Lastly, we forecast GHG emissions until 2050, reflecting changes in the population age structure. The age structure is expected to be impactful in countries where the proportion of elderly population will increase rapidly, such as Greece and South Korea. However, the average rate of reduction in OECD countries is expected to be only around 7% by 2050 (at an annual rate of 0.2%). This result shows that various policies, such as increasing taxes on fuel, improving fuel economy, and prioritizing fuel transition, should be applied to reduce GHG emissions. 

## Figures and Tables

**Figure 1 ijerph-17-07734-f001:**
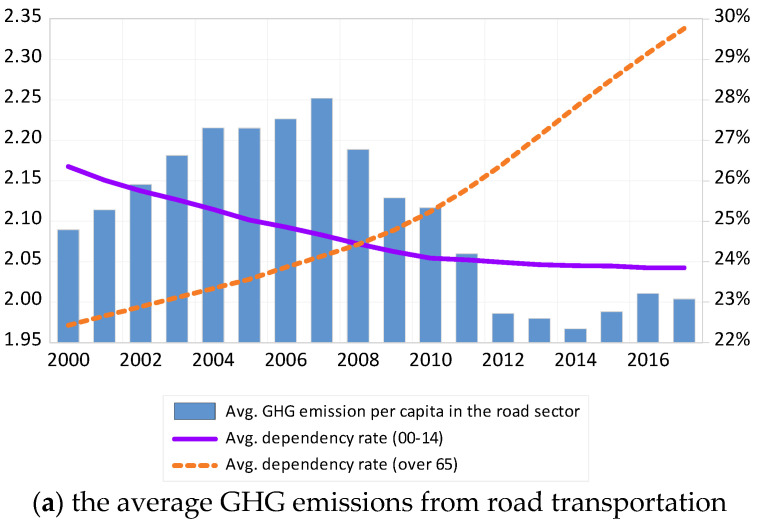
Age structure and GHG emissions.

**Figure 2 ijerph-17-07734-f002:**
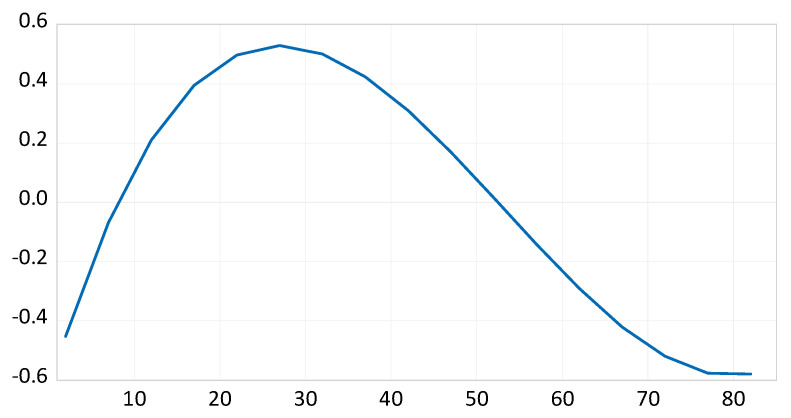
Implied coefficient of age structure: Model (1). Note: *x*-axis represents the age and *y*-axis represents the normalized effect.

**Figure 3 ijerph-17-07734-f003:**
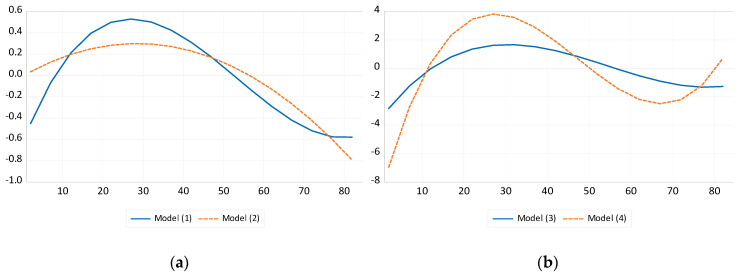
Comparison of age effect: Models (1)–(4). Note: *x*-axis represents the age and *y*-axis represents the normalized effect. (**a**) model (1) includes all the determinants of GHG emissions from road transportation; model (2) excludes the variable of polynomial structure. (**b**) model(3) excludes transport volume; model(4)uses only the factors related to personal transport use.

**Figure 4 ijerph-17-07734-f004:**
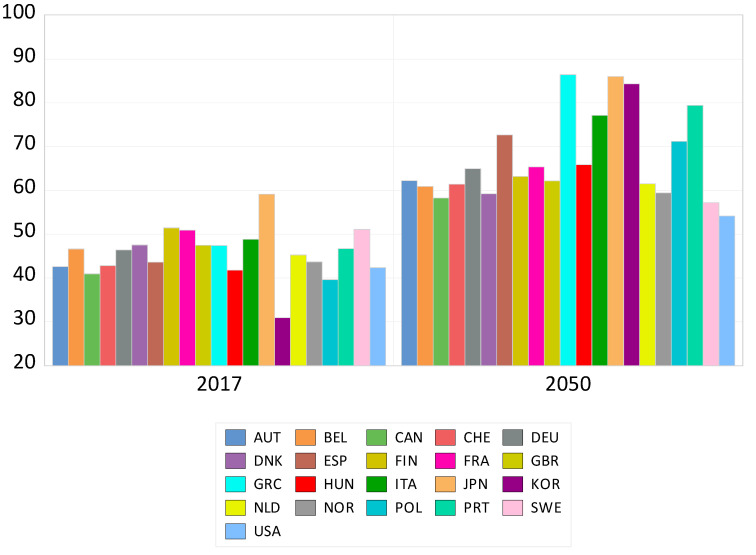
Change in the dependency rate of OECD countries (Unit: %). (Source: OECD Labor Force Statistics)

**Table 1 ijerph-17-07734-t001:** Variable descriptions.

Variable Name	Variable Definition	Unit	Data Source	Range
***E*** **(Emissions)**	Per capita greenhouse gas (GHG) emissions in road transportation	tCO_2_eq	Internationa Energy Agency (IEA), Enerdata	2000–2016 (annual)
***PR1–PR17***	Proportion of age groups (0–4, 5–9,…, 75–79, over 80)	%	World Bank, Organization for Economic Cooperation and Development (OECD)	
***Y*** **(Income)**	Per capita real gross domestic product (GDP)	2010 US$	World Bank	
***FP*** **(Fuel Price)**	Weighted average real price of gasoline and diesel (weighted as a proportion of gasoline and diesel consumption)	2015 US$/tonnes of oil equivalent (toe)	Adapted from Enerdata	
***VO*** **(Vehicle Ownership)**	Per capita number of registered vehicles	units/person	OECD	
***PD*** **(Population Density)**	Population density	person/km^2^	World Bank	
***UR*** **(Urbanization)**	Ratio of urban population	%	World Bank	
***FE*** **(Fuel Economy)**	(annual distance traveled × total registered vehicles)/total fuel consumption	km/L	Odyssee-Mure International Council for Clean Transportation (ICCT)	
***TV*** **(Transport Volume)**	Per capita transport volume of road transportation	million tons/person	OECD	
***FT*** **(fuel transition)**	Ratio of non-conventional fossil fuel consumption in road transportation (percentage of electricity, natural gas, and biofuel use)	%	IEA, Enerdata	

**Table 2 ijerph-17-07734-t002:** Descriptive statistics.

Variable	Mean	Maximum	Minimum	Standard Deviation	Observation
***E*** **(Emissions)**	2.1	5.3	0.7	0.9	357
***Y*** **(Income)**	41,655	91,566	8526	17,926	357
***FP*** **(Fuel Price)**	1660.7	2678.0	607.8	346.7	357
***VO*** **(Vehicle Ownership)**	0.6	0.9	0.3	0.1	352
***PD*** **(Population Density)**	169.0	525.4	3.4	145.3	357
***UR*** **(Urbanization)**	77.3	97.9	54.4	9.8	357
***FE*** **(Fuel Economy)**	14.0	23.8	9.5	2.2	342
***TV*** **(Transport Volume)**	3093	71,860	44	11,031	296
***FT*** **(Fuel Transition)**	2.7	16.4	0.0	2.8	357

**Table 3 ijerph-17-07734-t003:** Estimation Results: Models (1)–(4).

		Model (1) (Benchmark)	Model (2)	Model (3)	Model (4)
**Age structure effect**	*z1*	0.562 ***	0.119 **	2.256 ***	6.305 ***
	*z2* (×10)	−0.639 ***	−0.095 ***	−2.375 ***	−7.444 ***
	*z3* (×$10^{2}$)	0.189 **		0.689 ***	2.466 ***
**Personal transport use**	*Y*	0.642 ***	0.647 ***	0.842 ***	0.728 ***
	*FP*	−0.048 ***	−0.045 ***	−0.098 ***	−0.167 ***
	*VO*	0.252 ***	0.239 ***	0.358 ***	0.380 ***
**Macro transport** **environment**	*PD*	−0.891 ***	−0.920 ***	−1.004 ***	
	*UR*	−0.006 ***	−0.007 ***	−0.002 ***	
	*FE*	−0.075 ***	−0.068 ***	−0.248 ***	
	*TV*	0.113 ***	0.121 ***		
	*FT*	−0.009 ***	−0.010 ***		
**Wald test (H_0_: k = 2 vs. H_1_: k = 3)**		5.27 **	-	39.96 ***	310.11 ***
**Wald test (H_0_: k = 3 vs. H_1_: k = 4)**		0.83	-	263.34 ***	452.38 ***
**F-test of age structure**		140.19 ***	72.63 ***	165.11 ***	385.77 ***
**Obs.**		220	220	316	330
**Adj-R^2^**		0.984	0.984	0.985	0.977

Note: *** *p*-value < 0.01, ** *p*-value < 0.05. For the first Wald test, the null hypothesis is k = 2 and the alternative is k = 3. Also, for the second Wald test, the null is k = 3 and the alternative is k = 4.

**Table 4 ijerph-17-07734-t004:** Robustness estimation results: Robust (1) and Robust (2).

		Robust (1)	Robust (2)
**Age structure effect**	*PR0014*	−0.253	
	*PR65*	−1.378 ***	
	*DEPR0014*		−0.059 **
	*DEPR65*		−0.238 ***
**Personal transport use**	*Y*	0.645 ***	0.679 ***
	*FP*	−0.060 ***	−0.074 ***
	*VO*	0.244 ***	0.250 ***
**Macro transport** **environment**	*PD*	−1.028 ***	−1.062 ***
	*UR*	−0.006 ***	−0.005 ***
	*FE*	−0.014	−0.006
	*TV*	0.110 ***	0.100 ***
	*FT*	−0.009 ***	−0.009 ***
**F-test of age structure**		125.14 ***	84.98 ***
**Obs.**		220	220
**Adj-R^2^**		0.985	0.985

Note: *** *p*-value < 0.01, ** *p*-value < 0.05.

**Table 5 ijerph-17-07734-t005:** Forecast of GHG emissions per capita in road transport (Unit: tCO_2_eq).

	2010	2017	2030	2050	% Change (2017–2050)	Annual % Change (2017–2050)
**Austria**	2.60	2.67	2.57	2.50	−6.4%	−0.2%
**Belgium**	2.31	2.17	2.12	2.09	−3.6%	−0.1%
**Canada**	4.18	3.86	3.72	3.66	−5.2%	−0.2%
**Switzerland**	2.15	1.83	1.77	1.70	−6.9%	−0.2%
**Germany**	1.76	1.94	1.87	1.82	−6.0%	−0.2%
**Denmark**	2.20	1.94	1.89	1.87	−3.7%	−0.1%
**Spain**	1.85	1.76	1.69	1.61	−8.5%	−0.3%
**Finland**	2.20	1.97	1.93	1.86	−5.4%	−0.2%
**France**	1.83	1.82	1.77	1.76	−3.4%	−0.1%
**United Kingdom**	1.76	1.74	1.69	1.65	−5.2%	−0.2%
**Greece**	1.70	1.34	1.21	1.05	−21.7%	−0.7%
**Hungary**	1.14	1.28	1.23	1.19	−7.4%	−0.2%
**Italy**	1.75	1.52	1.46	1.40	−8.1%	−0.2%
**Japan**	1.57	1.45	1.40	1.34	−7.5%	−0.2%
**Korea, Rep.**	1.67	1.91	1.74	1.57	−17.9%	−0.6%
**Netherlands**	1.97	1.73	1.67	1.65	−4.8%	−0.1%
**Norway**	1.95	1.64	1.59	1.53	−6.6%	−0.2%
**Poland**	1.22	1.59	1.51	1.41	−11.2%	−0.4%
**Portugal**	1.67	1.54	1.47	1.39	−9.9%	−0.3%
**Sweden**	2.20	1.90	1.87	1.84	−2.8%	−0.1%
**United States**	4.76	4.48	4.36	4.28	−4.5%	−0.1%
**Average**	**2.12**	**2.00**	**1.93**	**1.87**	**−6.9%**	**−** **0.2%**

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
