# Peer review of "Population Age Structure and Greenhouse Gas Emissions from Road Transportation: A Panel Cointegration Analysis of 21 OECD Countries"

_ijerph, 2020, doi:10.3390/ijerph17217734_

Round 1
Reviewer 1 Report
Using panel model, this study attempted to analyze the relationship of age structure and GHG emission in 21 OECD countries. The article lacks a certain degree of innovation and fails to reflect the value of its research. it was not clear about what was the role of age structure in this paper. So the authors must comprehend the principle of the panel model, and analyze the date of 21 countries from the individual and time dimensions . However, it still needs some improvements which I try to list below:
- which effect model(fixed/random/mixed effect model) has been selected. Why?
- L230-L234. What is the relationship between z1, z2, z3 and Group 1, Group2, Group3. And also, What is the relationship between PR1~PR17 and z1, z2, z3.
- The authors selected 21 OECD countiesas their research object , but we could not find a separate discussion of each country or the whole research objects in section 4. Why?
- Please also carefully check with Citation styles throughout the paper.
Author Response
We appreciate your valuable comments.
|
# |
Comments |
Responses |
|
1 |
Overall comment The article lacks a certain degree of innovation and fails to reflect the value of its research. it was not clear about what was the role of age structure in this paper. So the authors must comprehend the principle of the panel model, and analyze the date of 21 countries from the individual and time dimensions |
We appreciate your valuable comment. Following your advice, we try to explain more on the implications and contributions of the paper. Previous studies only considered technical parts (such as income, fuel economy etc.) not the behavioral patterns of a driver. We think that age would affect driving pattern and transport demand. Hence, we try to analyze the relationship between age structure and transport emissions.
Those are well mentioned on row 53 to 66.
Also, this study deal with a panel model of 21 OECD countries. Hence, our results represent the overall relationship between age structure and the emission of developed countries. Hence, our results are not bounded to individual countries or time dimension but show the average pattern of developed countries. |
|
2 |
· which effect model(fixed/random/mixed effect model) has been selected. Why? |
In this study, we find a cointegration relationship between variables. It means that the variables have a long-run relationships, and in order to not miss long-run information, they should be modeled as a cointegration model. We use the FMOLS cointegration regression method in this analysis. This method is commonly used in the literature. |
|
3 |
· L230-L234. What is the relationship between z1, z2, z3 and Group 1, Group2, Group3. · · And also, What is the relationship between PR1~PR17 and z1, z2, z3.
|
Following your advice, we include a list of variables for each group in Table A.3 and footnote 4 to reduce misunderstanding. In footnote 4, we mention as follows: “Group 1 includes z1 and the other variables (E, Y, FP, VO, PD, UR, FE, TV, and FT). Group 2 includes z2 and the same other variables (E, Y, FP, VO, PD, UR, FE, TV, and FT). Finally, Group 3 includes z3 and the same other variables.”
We reorganize the result tables and rearrange them in the appendix section.
In the World Bank and OECD dataset, the age structure is divided into 17 age groups (PR1 ~ PR17). As mentioned in the paper on row 214-222, if we use all age groups together, it may cause three problems. Hence, we adjusted age groups into a polynomial function (the function of z variables). In this way, we can reduce the number of parameters to be estimated and ensures that the coefficients between age groups change smoothly. |
|
4 |
· The authors selected 21 OECD counties as their research object , but we could not find a separate discussion of each country or the whole research objects in section 4. Why?
|
Following your advice, we add the reason for selecting 21 OECD countries in the introduction section on row 41-45. We selected OECD countries since the GHG emission problem in the transportation sector is more serious in developed countries. Since our analysis is mainly about the OECD panel, our result and discussion is focused on the average effect of the age structure on OECD countries. However, we extend the analysis to make a projection of GHG emissions of individual countries up to 2050. From this analysis, we are able to compare countries’ age structure effect on GHG emissions. |
|
5 |
· Please also carefully check with Citation styles throughout the paper.
|
Following your advice, we adjusted overall citation styles. |
Reviewer 2 Report
Overall Comments
The topic is important. The manuscript is clearly written. The foundation and organization of the statistical analyses are well explicated.
The primary question is to what extent does the study provide new data (e.g., inverted-U shape effect of age structure on GHG emissions is not new)?
Recognizing the high quality of the analyses, however, what new insights for future scenarios are now possible that have not been feasible previously? The study stops short of reaching into the future.
For example, the global population is expected to increase to 11 billion by 2100. Given the projected age structure at 2100, what effect will the population have on GHG emissions? Or the increase in urban growth over the next decades that will occur primarily in developing countries?
The study would reach a wider readership if tCO2e data from transportation were included with the age cohorts cited, if this is feasible.
Suggested revisions
1) Place the purpose of the study in the first few lines of the abstract.
The effect of age and urban development on greenhouse gas (GHG) emissions from the transportation sector are critical for GHG emission reduction policies and for transportation industries. We address this problem using panel data of 21 OECD countries from 2000 to 2016 to analyze the effect of age structure on greenhouse gas (GHG) emissions in the road sector. By modelling the polynomial structure of age cohorts suggested by [1], we find an asymmetric inverted-U shaped effect of age structure on GHG emissions. People emit more GHGs as they age, and in their late 20s, people emit the maximum GHGs. After their mid-50s, people emit relatively less GHGs and after the age of 70, GHG reduction 18 becomes stagnant. In addition, personal factors such as income, fuel prices, and vehicle ownership have a significant effect on GHG emissions in the road sector. So too does the macro traffic environment factors (such as population density, urbanization, transport volume, and fuel transition). Our results extend previous findings by *[application to population structures in the future] and showing that the results hold true even when we consider other age structure variables, such as population ratio and dependency rate.
>>*What can you report that is new relative to previous findings?
2) Section 3. Summarize the statistical analysis in a brief introductory paragraph to assists readers in following your analysis.
3) Line 112--spell out the SIRPAT abbreviation.
4) Conclusion or discussion? The conclusion reads more like a repeat of statements previously made. A proper discussion will provide application of your insights to future scenarios of population growth and urban expansion.
5) What can the authors conclude about their study? Is the method now ready for integration into policy discussions? Is the method now reliable as a means of understanding urban growth and GHG emissions in a way that has not been contemplated?
Author Response
We appreciate your valuable comments.
|
# |
Comments |
Responses |
|
1 |
Overall comment The topic is important. The manuscript is clearly written. The foundation and organization of the statistical analyses are well explicated. The primary question is to what extent does the study provide new data (e.g., inverted-U shape effect of age structure on GHG emissions is not new)? Recognizing the high quality of the analyses, however, what new insights for future scenarios are now possible that have not been feasible previously? The study stops short of reaching into the future. For example, the global population is expected to increase to 11 billion by 2100. Given the projected age structure at 2100, what effect will the population have on GHG emissions? Or the increase in urban growth over the next decades that will occur primarily in developing countries? The study would reach a wider readership if tCO2e data from transportation were included with the age cohorts cited, if this is feasible.
|
We appreciate your valuable comment. Following your advice, we try to explain more on the implications and contributions of the paper. Previous studies only considered technical parts (such as income, fuel economy etc.) not the behavioral patterns of a driver. We think that age would affect driving pattern and transport demand. Hence, we try to analyze the relationship between age structure and transport emissions.
Those are well mentioned on row 53 to 66.
Further, following your advice, we made a projection of GHG emissions of 21 OECD countries up to 2050. We analyze the age structure effect by applying the projected demographic structure provided by OECD. The overall age structure effect is expected to be small, decreasing GHG emissions 0.2% annually. From this projection, we can conclude that to achieve the Paris Agreement goal, we should implement measures such as enhancing fuel efficiency and transport infrastructure together.
The projection result is analyzed in Section 4.4 on row 387-420 |
|
2 |
Place the purpose of the study in the first few lines of the abstract. >>*What can you report that is new relative to previous findings? |
Following your advice, we include a short contribution of our paper in the abstract section. |
|
3 |
Section 3. Summarize the statistical analysis in a brief introductory paragraph to assists readers in following your analysis. |
Following your advice, we include an introductory paragraph on row 191-195. |
|
4 |
Line 112--spell out the SIRPAT abbreviation. |
We added the full name of STIRPAT model on row 53.
STIRPAT stands for STochastic Impacts by Regression on Population, Affluence and Technology |
|
5 |
Conclusion or discussion? The conclusion reads more like a repeat of statements previously made. A proper discussion will provide application of your insights to future scenarios of population growth and urban expansion. |
Following your advice, we add a discussion section and suggest policy implications in the section.
Discussion section is on row 423-434. |
|
6 |
What can the authors conclude about their study? Is the method now ready for integration into policy discussions? Is the method now reliable as a means of understanding urban growth and GHG emissions in a way that has not been contemplated? |
From the result of 4.3, we can conclude that the age structure may reduce GHG emissions in the transport sector, but the overall effect is very small (decreasing 0.2% annually). Hence, to achieve sustainable transport emission, OECD countries should implement efficient policy measures, such as enhancing fuel efficiency and transport infrastructure. |
Reviewer 3 Report
The introduction and literature review are very poorly written. It reads like a note of literatures rather than coherent writing that weaves the related literatures together that leads to the authors’ work. Also, there are many weird short paragraphs that should be avoided. And due to the poor setup, the paper reads more like a technical report than an academic paper. It is hard to read the policy question and broader impact from the paper. It also needs a discussion section for these questions. For example, how age, income interplay with factors such as built-environment and human behaviors to influence transportation GHG emissions? https://journals.sagepub.com/doi/abs/10.1177/2399808320924437 https://www.sciencedirect.com/science/article/pii/S0921800913000980 https://www.sciencedirect.com/science/article/pii/S0306261916314957 https://www.tandfonline.com/doi/abs/10.1080/00343404.2019.1701186 https://www.sciencedirect.com/science/article/pii/S0966692316305981 The method description is very sloppy. There is no identification strategy on how age cohort is determined (actual drivers or neighborhood composition). Title: It is unclear whether “Age structure” means population age or road age. Please specify. Abstract Line 16: Remove citation in the abstract. Explain the key background by words. Abstract line 22: Again, previous literature should not appear in abstract, because the authors cannot assume all readers to be familiar with these literatures. Please be descriptive and self-explanatory in the abstract. Line 27: Numbers need references. Line 34: These very short paragraphs are weird. Line 39: “Only a few studies…” does not show your innovation. Please specify how your study contributes to previous findings and what is the innovation. Line 57: Again, avoid very short paragraphs. Section 2.3: Describing the work does not show contribution. Contribution should address the shortcoming in previous research, Line 147: What literature? Not all works in the literature review use the same model. Please specify. How do the authors determine the cohort? Are they actually drivers from travel survey, or age group composition from the census data? Section 4.1. Though the tests are necessary, none of the result tables show significant information interests. I suggest relegating them to appendix and only show test conclusions in the main text. Line 272: Variable explanations should be conducted in the data section not in the result section. Figure 3: The axis labels are missing.Author Response
We appreciate your valuable comments
|
# |
Comments |
Responses |
|
1 |
The introduction and literature review are very poorly written. It reads like a note of literatures rather than coherent writing that weaves the related literatures together that leads to the authors’ work. Also, there are many weird short paragraphs that should be avoided. And due to the poor setup, the paper reads more like a technical report than an academic paper. It is hard to read the policy question and broader impact from the paper. It also needs a discussion section for these questions. For example, how age, income interplay with factors such as built-environment and human behaviors to influence transportation GHG emissions? |
We appreciate your valuable comment. We understand that our motivation is not well presented in the paper. Following your advice, we try to explain more on the implications and contributions of the paper. Previous studies only considered technical parts (such as income, fuel economy etc.) not the behavioral patterns of a driver. We think that age would affect driving pattern and transport demand. Hence, we try to analyze the relationship between age structure and transport emissions.
Those are well mentioned on row 53 to 66.
Further, we avoid short paragraphs and try to reorganize literature review section.
|
|
2 |
The method description is very sloppy. There is no identification strategy on how age cohort is determined (actual drivers or neighborhood composition) |
We understand that the derivation process in Section 3.1 is rather tedious and sloppy. Following your advice, we include an introductory paragraph on row 191-195 to help understanding and explain the process of derivation.
Further, the explanation about the age groups we used is onn row 234-239. In this study, we use the most disaggregated age structure information provided by WB and OECD. |
|
3 |
Title: It is unclear whether “Age structure” means population age or road age. Please specify. |
Following your advice, we change the title to ‘Population age structure’. Further, we change the word ‘cohort’ to ‘age group’ to avoid misunderstanding. |
|
4 |
Abstract Line 16: Remove citation in the abstract. Explain the key background by words. Abstract line 22: Again, previous literature should not appear in abstract, because the authors cannot assume all readers to be familiar with these literatures. Please be descriptive and self-explanatory in the abstract |
Following your advice, we remove citation in the abstract and try to clarify the abstract section
|
|
5 |
Line 27: Numbers need references |
We add references for statistics. The number you mentioned is in row 38. We include the source of the data. |
|
6 |
Line 34: These very short paragraphs are weird |
Following your advice, we avoid short paragraphs |
|
|
. Line 39: “Only a few studies…” does not show your innovation. Please specify how your study contributes to previous findings and what is the innovation. |
Following your advice, we try to explain contributions of our paper in the end of introduction section on row 67-93. |
|
|
Line 57: Again, avoid very short paragraphs. |
We adjusted short paragraphs |
|
|
Section 2.3: Describing the work does not show contribution. Contribution should address the shortcoming in previous research |
Following your advice, we try to explain contributions of our paper in the end of introduction section on row 67 to 93. Further, we try to mention the shortcomings of previous studies and our contributions. |
|
|
Line 147: What literature? Not all works in the literature review use the same model. Please specify. |
We include references for the literature. |
|
|
How do the authors determine the cohort? Are they actually drivers from travel survey, or age group composition from the census data? |
We used the most disaggregated age group data provided by the World Bank and OECD. This is mentioned in Section 3.1 (row234-239) Also, age is known to be an important factor for driving patterns which affects GHG emissions in the transport sector. Literature reviews about the survey data is mentioned on row 59-66 and Section 2.2 (row143-181)
|
|
|
Section 4.1. Though the tests are necessary, none of the result tables show significant information interests. I suggest relegating them to appendix and only show test conclusions in the main text. |
Following your advice, we rearrange unit root and cointegration test results in the appendix section |
|
|
Line 272: Variable explanations should be conducted in the data section not in the result section |
Following your advice, we delete variable explanations in the result section. |
|
|
Figure 3: The axis labels are missing. |
Following your advice, we add a note for explaining labels below the graph.
“ Note. x-axis represents the age and y-axis represents the normalized effect.” |
Round 2
Reviewer 3 Report
The revision has addressed reviewer concerns and improved the qualityof the manuscript. However, the writing and language is still sloppy and could use some language editing services.
Author Response
While writing our manuscript, we were cautious not to write unprofessional English. Hence, we used language editing services twice and made corrections for the manuscript. The following pictures show the certificate of the English editing procedure. Panel (A) shows the first editing service we went through before submitting the manuscript. However, we found that the first editing was not professional enough. Hence, we asked for more elaborated English editing after revising the comments from the first round of peer review. Panel (B) is the certificate of the second editing. During the second editing process, we tried to change the structure of sentences and paragraphs to clear our argument. The editing service also provided adjustment of styles and tone of the manuscript.
Following your advice, we reviewed the paper ourselves and try our best to correct the language and enhance the explanatory power.
